# Unexpected enhancement of new particle formation by lactic acid

#### sulfate resulting from SO<sub>3</sub> loss in forested and agricultural regions 2

3 Rui Wang, Shuqin Wei<sup>‡</sup>, Zeyao Li<sup>‡</sup>, Kaiyu Xue, Rui Bai, Tianlei Zhang\*

4 Shaanxi Key Laboratory of Catalysis, School of Chemical & Environment Science, Shaanxi

5 University of Technology, Hanzhong, Shaanxi 723001, P. R. China

#### 6 Abstract

1

7

8

11

13

Organosulfates (OSs) are key components of atmospheric aerosols and serve as tracers for secondary organic aerosol (SOA) formation. Among these, lactic acid sulfate (LAS) has been 9 increasingly detected in the atmosphere. However, its molecular formation pathways and its role in 10 new particle formation (NPF) remain poorly understood. In this work, we investigate the gas-phase formation mechanism of LAS via the reaction between lactic acid (LA) and SO3, and assess its 12 impact on sulfuric acid-ammonia (SA-A) driven NPF using quantum chemical calculations and Atmospheric Cluster Dynamics Code (ACDC) kinetic modeling. Our results show that SA and H<sub>2</sub>O 14 significantly catalyze the LA-SO<sub>3</sub> reaction, enhancing the effective rate coefficient by 7-10 orders of magnitude within the temperature range of 280-320 K. Further molecular-level analysis using the ACDC reveals that LAS not only significantly enhances the clustering stability of SA and A up to 108-fold, but also plays a significant and direct role in SA-A nucleation under conditions typical of forested and agricultural regions. Notably, LAS-SA-A clusters contribute to 97% of the overall cluster formation pathways in regions with high LAS concentrations like Centreville, Alabama. Additionally, our findings show that the nucleation potential of LAS-SA-A clusters is stronger than that of LA-SA-A clusters, aligning with field observations, even though LAS concentrations are typically three orders of magnitude lower than LA. These findings imply that OSs formed through SO<sub>3</sub> consumption may significantly contribute to the enhanced NPF rates observed in continental regions.

<sup>\*</sup> Corresponding authorsTel: +86-0916-2641083, Fax: +86-0916-2641083. E-mail addresses: ztianlei88@163.com (T. L Zhang)

<sup>\*</sup> Shuqin Wei and Zeyao Li contributed equally to this work.

2627

47

49

52

## 1 Introduction

Atmospheric aerosol particles pose significant risks to public health, adversely affecting both the respiratory and cardiovascular systems (Anderson et al., 2012; Xing et al., 2016; Zhang et al., 2023b). Beyond health implications, these particles contribute to global warming by reducing visibility and disrupting the Earth's radiative balance (Lund et al., 2019; Zheng et al., 2018). As a major source of atmospheric aerosols, new particle formation (NPF), accounts for over 50% of the total particle number concentration and is strongly associated with severe haze events in megacities across China (Kulmala et al., 2004; Brean et al., 2020). Despite its significance, accurately characterizing the NPF process remains a considerable challenge, primarily due to limitations in current measurement techniques and an incomplete comprehension of the underlying mechanisms. While field observations and CLOUD chamber experiments (Kulmala et al., 2004; Dai et al., 2023; Lee et al., 2019; Hirsikko et al., 2011; Zhang et al., 2015) have provided valuable insights, they are insufficient to fully elucidate these processes. To address these gaps, a molecular-level approach is essential, as it allows for a more precise understanding of nucleation mechanisms (Yang et al., 2021; Li et al., 2017). This approach enables the detailed determination of molecular cluster geometries, the strengths of intermolecular interactions, and the pathways of cluster formation (Long et al., 2013; Zu et al., 2024b; Rong et al., 2020b). Such molecular insights are critical to evaluating the impacts of aerosols on the atmosphere and for devising effective strategies to mitigate haze formation. Gaseous sulfuric acid (SA), derived from the oxidation of SO<sub>2</sub>, has long been recognized as a key NPF precursor (Kirkby et al., 2011; Zhao et al., 2024). Molecular-level studies have shown that various nucleation precursors, including water (H2O) (Zhang et al., 2012b), ammonia (NH3) (Kirkby et al., 2011; Zhang et al., 2015), methylamine (MA) (Shen et al., 2020), dimethylamine (DMA) (Cai et al., 2021; Kurtén et al., 2008), monoethanolamine (MEA) (Shen et al., 2019), piperazine (PZ) (Ma et al., 2019) and iodic acid (Sipilä et al., 2016), are involved in SA-driven binary nucleation, which serves as a primary initiator of NPF. However, binary nucleation mechanisms alone cannot fully account for the discrepancies observed between measured and modeled global NPF rates (Hodshire et al., 2019; Kirkby et al., 2016), suggesting the involvement of additional gaseous species. In response to this, several studies have explored the role of ternary nucleation driven by SA-A, which involves a broader array of atmospheric species, including ammonia (NH<sub>3</sub>) (Li et al.,

2020b; Yin et al., 2021a), organic amines (Li et al., 2017; Li et al., 2018a), organic and inorganic 55 acids (Wang et al., 2011; Liu et al., 2018), and highly oxidized multifunctional compounds (HOMs) (Liu et al., 2021a; Liu et al., 2019a; Ning and Zhang, 2022; Yin et al., 2021b; Zhang et al., 2018). 56 57 Despite recognizing the enhancement provided by SA-A-driven ternary nucleation, the nucleation 58 rates predicted by these mechanisms still fall short when compared to field observations (Kirkby et al., 2016; Hodshire et al., 2019; Yin et al., 2021a). The persistent underestimation underscores the 59 60 need for further investigation into the role of additional gaseous species to better understand the complex mechanisms driving NPF. 61 62 Organosulfates (OSs), formed through the chemical transformation of organic acids, constitute a major portion of organosulfur species in atmospheric aerosols, contributing 5-30% to the organic 63 64 mass in PM10 (Sun et al., 2025; Brüggemann et al., 2017). These compounds are prevalent in 65 atmospheric particles and are commonly employed as markers to track the formation of secondary 66 organic aerosols (SOAs) in environmental research (Tan et al., 2022; Zhang et al., 2012a; Froyd et 67 al., 2010a; Brüggemann et al., 2017; Mutzel et al., 2015; Glasius et al., 2017). Recent research has 68 led to the identification and characterization of various OSs in fine particulate matter samples from 69 regions including the United States, China, Mexico City and Pakistan (Hettiyadura et al., 2017; 70 Wang et al., 2018; Olson et al., 2011). Meanwhile, studies suggest that the cycloaddition of SO<sub>3</sub> to 71 organic acids could be a key mechanism for OSs formation resulting in compounds with lower vapor 72 pressures than their parent carboxylic acids and increased inter-molecular interaction sites (Smith 73 et al., 2020; Tan et al., 2020; Yao et al., 2020; Zhang et al., 2023a). Notably, lactic acid sulfate (LAS) 74 has been identified as the dominant OSs species across all these field observations (Darer et al., 75 2011; Riva et al., 2015; Kundu et al., 2013). However, the specific formation mechanism of LAS 76 from the reaction of lactic acid (LA) with SO<sub>3</sub> remains largely unexplored. Additionally, SA and 77 water (H<sub>2</sub>O) (Tan et al., 2022; Zhang et al., 2025; Li et al., 2018b), both prevalent in the atmosphere, 78 act as strong hydrogen atom donors/acceptors, facilitating proton transfer reactions and potentially 79 catalyzing the LA-SO3 reaction. 80 The reaction products of SO<sub>3</sub> with major atmospheric trace species have been shown proven 81 to significantly influence the formation of NPF. For instance, compounds such as sulfamic acid (Li 82 et al., 2018a), oxalic sulfuric anhydride (Yang et al., 2021), methyl hydrogen sulfate (Liu et al., 83 2019a), glyoxylic sulfuric anhydride (Rong et al., 2020a) and formic acid sulfate (Wang et al., 2025),

85

8687

95

generated through reactions of SO<sub>3</sub> with ammonia, oxalic acid, methanol, glyoxylic acid and formic acid, all exhibit catalytic effects on NPF in aerosols. Structurally, LAS, the product of the SO<sub>3</sub> + LA reaction, contains both -COOH and -SO<sub>3</sub>H functional groups, which facilitate additional hydrogen bonding with atmospheric particle precursors (Yao et al., 2020). However, the role of LAS in enhancing SA-A nucleation remains underexplored, limiting our ability to comprehensively evaluate its impact on NPF processes. Furthermore, LA, a highly oxidized α-hydroxy acid with both -OH and -COOH groups (Mochizuki et al., 2019), can enhance the stability of SA-A clusters and facilitate NPF (Li et al., 2017). Given its relatively larger atmospheric concentrations, particularly in regions with elevated organic acid pollution, LA may also significantly influence NPF. So, understanding the distinct contributions of LAS and LA to SA-A nucleation is crucial, as this will advance our understanding of NPF events, particularly in agricultural and forested regions. In this work, we utilized quantum chemical calculations together with master equation analysis to investigate the gas-phase reaction of SO3 with LA that forms LAS, with H2O and SA serving as catalysts. The role of LAS in enhancing SA-A nucleation was then explored by examining the formation mechanisms of the (LAS)<sub>x</sub>(SA)<sub>y</sub>(A)<sub>z</sub> ( $0 \le z \le x + y \le 3$ ) system using the Atmospheric Clusters Dynamic Code (ACDC) kinetic model. Additionally, the potential influence of LAS on atmospheric nucleation and particle formation (NPF) was assessed across diverse global regions. Finally, a comparative study of LA and LAS was also conducted to elucidate the respective roles of organic acids and OSs in enhancing SA-A nucleation, focusing on the formation mechanisms of both LA-SA-A and LAS-SA-A systems.

## 2 Methodology

#### 2.1 Quantum chemical calculations

The gas-phase reaction of SO<sub>3</sub> with LA to form LAS, both in the absence and presence of H<sub>2</sub>O and SA as catalysts, was systematically optimized and calculated using the Gaussian 09 program (Faloona et al., 2009) at the M06-2X/6-311++G(2df,2pd) level (Stewart, 2007; Walker et al., 2013). Intrinsic reaction coordinate analyses (Hratchian and Schlegel, 2005) were carried out at the same computational level to verify the connection between transition states and their respective pre-reactive complexes and products. Furthermore, single-point energy calculations were refined at the CCSD(T)-F12/cc-pVDZ-F12 level with the ORCA program (Neese, 2012),

employing the optimized geometries as input.

To identity the global minimum energy configurations of  $(SA)_x(A)_y(LAS)_z$  clusters ( where  $0 \le y \le x + z \le 3$ ), we utilized the ABCluster program (Zhang and Dolg, 2016) to systematically generate initial structures for various clusters combinations. Specifically, using the ABCluster procedure and the CHARMM force field, a diverse set of initial structures  $n \times 1000$  ( $1 < n \le 4$ ) were randomly produced. Initially, the primary structures were optimized and their energies were ranked using the PM6 method in MOPAC 2016 (Partanen et al., 2016; Stewart, 2007). After the initial sampling, considering the excellent performance of the M06-2X method in accurately characterizing the geometries of atmospheric clusters (Walker et al., 2013; Lu et al., 2020), up to 1000 favorable configurations were selected for rigorous re-optimization at the M06-2X/3-21G\* level of theory. Subsequently, the 100 lowest-energy configurations were further optimized using the M06-2X/6-31G(d, p) level of theory, from which the 10 configurations with the lowest energies were identified. Finally, to accurately determine the global minimum, the M06-2X/6-311++G(2df, 2pd) method was applied to refine these 10 lowest-energy configurations.

## 2.2 Rate coefficients calculations

Rate constants for the SO<sub>3</sub> + LA reaction, both without and with H<sub>2</sub>O and H<sub>2</sub>SO<sub>4</sub> as catalysts, were determined via Rice-Ramsperger-Kassel-Marcus (RRKM) theory (Glowacki et al., 2012; Wardlaw and Marcus, 1984) within the Master Equation (ME/RRKM) framework in MESMER (Master Equation Solver for Multi-Energy Well Reactions) code (Glowacki et al., 2012; Klippenstein and Marcus, 1988). Specifically, in the MESMER calculations, the rate constants for the barrierless formation of pre-reactive complexes from reactants were determined using the Inverse Laplace Transform (ILT) method (Horváth et al., 2020), whereas the subsequent conversion of these complexes to products via transition states was evaluated using RRKM theory (Mai et al., 2018). The ILT method and RRKM theory can be represented in Eqs. (1) and (2), respectively:

$$k^{\infty}(\beta) = \frac{1}{Q(\beta)} \int_{0}^{\infty} k(E) \rho(E) \exp(-\beta E) dE$$
(1)

$$k(E) = \frac{W(E - E_0)}{h\rho(E)}$$
(2)

Here, h represents Planck's constant,  $\rho(E)$  indicates the density of accessible states for the reactant at energy E,  $E_0$  is the reaction threshold energy and  $W(E-E_0)$  refers to the rovibrational states of the

152

159

160

161162

163164

165

166167

168

- transition state, excluding motion along the reaction coordinate. Geometries, vibrational frequencies,
- and rotational constants were obtained at the M06-2X/6-311++G(2df,2pd) level, with single-point
- energies refined at the method of CCSD(T)-F12/cc-pVDZ-F12.

#### 2.3 ACDC kinetics simulation

- The ACDC was utilized to investigate the molecular-level collision coefficient ( $\beta$ , cm<sup>3</sup> s<sup>-1</sup>), evaporation coefficient ( $\gamma$ , s<sup>-1</sup>) and cluster formation rates (J, cm<sup>-3</sup> s<sup>-1</sup>). Thermodynamic parameters and structural information for cluster formation, obtained from quantum chemical calculations performed by M06-2X/6-311++G(2 $d\gamma$ , served as input parameters for the ACDC model. The MATLAB-R2014a platform, leveraging its odel5s solver (Shampine and Reichelt, 1997), performed numerical integration of the birthdeath equation for the ACDC model, thereby elucidating the kinetics of cluster growth over time. The general form of the birth-death equation for the
- $\frac{dc_i}{dt} = \frac{1}{2} \sum_{i < i} \beta_{j,(i-j)} c_j c_{(i-j)} + \sum_i \gamma_{(i+j) \to i} c_{i+j} \sum_j \beta_{i,j} c_i c_j \frac{1}{2} \sum_{j < i} \gamma_{i \to j} c_i + Q_i S_i \quad (3)$
- In this formulation,  $\beta_{i,i}$  corresponds to the collision frequency factor between clusters of sizes i and
- $j, \gamma_{(i+j)\rightarrow i}$  quantifies the fragmentation rate of composite clusters into their constituent monomers i
- and j. The system's open nature is accounted for through  $Q_i$ , representing the external flux of cluster
- i, and  $S_i$ , characterizing its removal rate. Here, the condensation sink coefficient was assigned 2.6 ×
- 10<sup>-3</sup> (Liu et al., 2021b).

## 3 Results and discussions

concentration  $c_i$  of cluster i given by,

## 3.1 Formation of LAS via the reaction of SO<sub>3</sub> with LA

In the direct cycloaddition pathway (Channel LAS) illustrated in Fig. 1, the hydroxyl (-OH) group of LA reacts with the sulfur atom of  $SO_3$ , leading to the formation of LAS via proton transfer from LA to  $SO_3$ . However, the resulting  $SO_3$ ···LA complex (denoted as IM) is thermodynamically unstable, primarily due to the significant ring strain in the four-membered structure, exhibiting a relative Gibbs free energy of 5.6 kcal·mol<sup>-1</sup>. The Gibbs free energy barrier for this reaction is calculated to be 22.3 kcal·mol<sup>-1</sup>. As indicated in Table S5, the rate coefficients for Channel LAS are extremely low, spanning from  $1.35 \times 10^{-26}$  to  $6.21 \times 10^{-25}$  cm<sup>3</sup>·molecule<sup>-1</sup>·s<sup>-1</sup> across the temperature range of 230-320 K. These values suggest that this pathway is both slow and thermodynamically

169 unfavorable for LAS formation under typical atmospheric conditions. 170 H<sub>2</sub>O, highly abundant in the atmosphere with concentration around 10<sup>17</sup> molecules cm<sup>-3</sup> (Huang et al., 2015; Tan et al., 2022), serves as both a donor and acceptor of hydrogen bonds, and 171 172 is widely recognized for its ability to catalyze a wide range of proton transfer reactions. To assess 173 its catalytic effect on the formation of LAS, we examined the SO<sub>3</sub> + LA reaction in the presence of 174 H<sub>2</sub>O (Channel WM), as illustrated in Fig. 1. This reaction can proceed via three possible sequential 175 bimolecular pathways: (i) SO<sub>3</sub>···LA + H<sub>2</sub>O, (ii) SO<sub>3</sub>···H<sub>2</sub>O + LA and (iii) LA···H<sub>2</sub>O + SO<sub>3</sub>. Considering typical atmospheric concentrations of SO<sub>3</sub> (10<sup>5</sup> molecules cm<sup>-3</sup>) (Zhang et al., 2024), 176 LA (10<sup>12</sup> molecules cm<sup>-3</sup>) (Li et al., 2017) and H<sub>2</sub>O (10<sup>17</sup> molecules cm<sup>-3</sup>) (Huang et al., 2015; Tan 177 178 et al., 2022), the calculated concentrations of SO<sub>3</sub>···LA, SO<sub>3</sub>···H<sub>2</sub>O and LA···H<sub>2</sub>O complexes at 298 K are  $4.18 \times 10^{-2}$ ,  $5.80 \times 10^{3}$  and  $2.32 \times 10^{8}$  molecules cm<sup>-3</sup>, respectively (see Table S2 in 179 180 the Supplement). These results suggest that Channel WM predominantly proceeds via the collision of LA···H<sub>2</sub>O with SO<sub>3</sub>. 181 182 The free energy barrier for Channel WM is 7.8 kcal·mol<sup>-1</sup>, which is 14.5 kcal·mol<sup>-1</sup> lower than 183 the barrier for the uncatalyzed cycloaddition pathway. At the experimental concentration of H<sub>2</sub>O  $([H_2O] = 10^{17} \text{ molecules} \cdot \text{cm}^{-3})$  (Huang et al., 2015; Tan et al., 2022), the effective rate coefficient 184 for the H<sub>2</sub>O-catalyzed reaction is 2.00 × 10<sup>-16</sup> cm<sup>3</sup> molecule<sup>-1</sup> s<sup>-1</sup>, which is nine orders of magnitude 185 186 greater than the rate for the direct cycloaddition pathway (2.22 × 10<sup>-25</sup> cm<sup>3</sup> molecule<sup>-1</sup> s<sup>-1</sup>). These 187 results clearly demonstrate that the H<sub>2</sub>O-catalyzed LA + SO<sub>3</sub> reaction represents a significantly more 188 favorable route for LAS formation. Detailed effective rate coefficients for the H<sub>2</sub>O-catalyzed 189 reaction are provided in Fig. 2(a). 190 SA is another abundant atmospheric species that efficiently donates and accepts hydrogen bond, 191 facilitating proton transfer (Yao et al., 2018; Tan et al., 2018) and potentially catalyzing the LA + 192 SO<sub>3</sub> reaction. As shown in Fig. 1, SA is significantly more effective than H<sub>2</sub>O in promoting LAS 193 formation via cycloaddition. Specifically, SA increases the stabilization energy of the SO<sub>3</sub>···LA 194 complex by 7.1 kcal·mol<sup>-1</sup>, 5.0 kcal·mol<sup>-1</sup> greater than the stabilization provided by H<sub>2</sub>O and reduces 195 the distance between the oxygen atom of the -OH group in LA and the sulfur atom in SO<sub>3</sub> by 0.09 Å in the SO<sub>3</sub>···LA···SA complex. Thermodynamically, SA lowers the Gibbs free energy barrier to 196 3.5 kcal·mol<sup>-1</sup>, 4.3 kcal·mol<sup>-1</sup> lower than the barrier observed for the H<sub>2</sub>O-catalyzed pathway. The 197 198 effective rate coefficients for the SA ([SA] =  $10^7$  molecules cm<sup>-3</sup>)-catalyzed reaction ( $k'_{SA}$ ) is 4-5

228

humidity, indicating that SA is kinetically more favorable, particularly at altitudes of 5-10 km. Thus, SA predominantly catalyzes the SO<sub>3</sub> + LA reaction, significantly contributing to the gas-phase loss 201 202 of SO<sub>3</sub> in LA-rich atmospheric regions. 203 Previous theoretical studies have indicated that atmospheric acids can catalyze the hydrolysis 204 of SO<sub>3</sub> to form SA (Hazra and Sinha, 2011; Cheng et al., 2022; Long et al., 2013; Lv et al., 2019). 205 In this context, the potential catalytic role of LA in SO<sub>3</sub> hydrolysis was also explored. The potential 206 energy surface (PES) for this reaction is presented in Fig. S1, with the effective rate coefficients 207 compared to those for SO<sub>3</sub> hydrolysis catalyzed by SA, HNO<sub>3</sub>, HCOOH, and OA. As shown in Fig. 208 2(b), LA predominantly catalyzes SO<sub>3</sub> hydrolysis within the temperature range of 280-320 K at a 209 concentration of  $1.0 \times 10^{12}$  molecules cm<sup>-3</sup>. Besides, given the current lack of atmospheric field data 210 on gas-phase LAS and lactic acid sulfuric anhydride (LASA), thermodynamic equilibrium 211 calculations were used to estimate their concentrations and assess their potential impacts on atmospheric NPF. Modeling results suggest LAS concentrations of 10<sup>3</sup>-10<sup>5</sup> molecules cm<sup>-3</sup>, which 212 is nine orders of magnitude higher than that of LASA (ranging from 10<sup>-6</sup>-10<sup>-4</sup> molecules · cm<sup>-3</sup>). This 213 214 suggests that LAS has significantly more atmospheric relevance than LASA, with a correspondingly 215 higher potential to influence NPF. Detailed calculations and further insights are provided in Table 216 217 3.2 Enhancing effect of LAS on SA-A-driven NPF 218 The role of LAS in promoting SA-A-driven NPF process was thoroughly examined. Initially, 219 potential interaction sites between LAS and SA-A clusters were identified through molecular 220 analyses. Next, the stable structures and thermodynamic stabilities of various  $(LAS)_x(SA)_y(A)_z$  ( $y \le$ 221  $x + z \le 3$ ) clusters were characterized, providing insight into their structural integrity. Building on 222 these findings, the nucleation mechanism of the SA-A-LAS system was investigated, with a 223 particular focus on the impact of temperature and precursor concentrations on LAS-mediated NPF processes. Finally, the atmospheric implications of LAS-enhanced SA-A nucleation were evaluated, 224 225 especially in forested and agricultural-developed regions. 226 3.2.1 Cluster thermodynamic data 227 Stable cluster formation is primarily driven by strong interactions between nucleation

orders of magnitude higher than that for the H<sub>2</sub>O-catalyzed pathway (k'<sub>WM</sub>) at 100 % relative

precursors (Lu and Chen, 2012). To assess the binding potential of LAS with the SA-A cluster, the

229 electrostatic potential (ESP)-mapped molecular van der Waals surface was calculated to identify 230 key interaction sites. As shown in Fig. 3, the hydrogen atom of the -SO<sub>3</sub>H moiety in LAS exhibits 231 a positive ESP of +78.73 kcal·mol<sup>-1</sup>, suggesting its role as a hydrogen bond donor that can interact 232 with the double-bonded oxygen atom of SA or the nitrogen atom of A, both of which act as hydrogen 233 bond acceptors. Additionally, the double-bonded oxygen in LAS, with a negative ESP of -32.51 234 kcal·mol<sup>-1</sup>, can act as a hydrogen-bond acceptor, interacting with the hydroxyl hydrogen of SA (-235 OH) or the hydrogen of A. These intermolecular interactions imply that LAS enhances nucleation efficiency between SA and A during aerosol nucleation, thereby stabilizing the resulting molecular 236 237 clusters. Based on the ESP analysis, the most stable configurations of  $(LAS)_x(SA)_y(A)_z$  ( $z \le x + y \le$ 238 3) clusters were identified (Fig. S2), with the observed interaction sites in the ternary clusters 239 corresponding well to the ESP predictions. 240 To quantitatively evaluate the binding strength of LAS within binary SA-A-based clusters, the Gibbs free energies ( $\Delta G$ , kcal·mol<sup>-1</sup>, Table S7) for the (LAS)<sub>x</sub>(SA)<sub>y</sub>(A)<sub>z</sub> ( $z \le x + y \le 3$ ) clusters were 241 242 calculated at temperatures of 238.15 K, 258.15 K, 278.15 K and 298.15 K. All clusters exhibited 243 negative  $\Delta G$  values, confirming thermodynamic favorability. Importantly, ternary SA-A-LAS 244 clusters consistently demonstrated lower  $\Delta G$  values compared to their binary counterparts, 245 suggesting that the presence of LAS reinforces the stability of SA-A clusters. Further analysis of 246 stability at 278.15 K was carried out by examining total evaporation rates ( $\Sigma \gamma$ ), derived from cluster 247  $\Delta G$  values (Table S7) and collision rates ( $\beta$ , Table S8), as summarized in Fig. 4. Previous research 248 indicates that lower  $\sum \gamma$  are indicative of greater cluster stability (Li et al., 2024; Zu et al., 2024a). 249 At 278.15 K, clusters incorporating LAS exhibit a lower  $\sum \gamma$  compared to those composed solely of SA and A molecules. For example, the  $\Sigma \gamma$  values for the (A)<sub>1</sub> (LAS)<sub>1</sub> (1.19 × 10<sup>4</sup> s<sup>-1</sup>) and 250 251  $(A)_3 \cdot (LAS)_3 \cdot (8.64 \times 10^{-8} \text{ s}^{-1})$  clusters were  $3.1 \cdot 10^8$  times lower than those for the  $(SA)_1 \cdot (A)_1 \cdot (3.73)_1 \cdot (A)_2 \cdot (A)_3 \cdot (A)_3 \cdot (A)_4 \cdot (A)_4 \cdot (A)_4 \cdot (A)_4 \cdot (A)_5 \cdot (A)_5 \cdot (A)_6 \cdot (A)_6$  $\times$  10<sup>4</sup> s<sup>-1</sup>) and (SA)<sub>3</sub>·(A)<sub>3</sub> (3.28  $\times$  10<sup>1</sup> s<sup>-1</sup>) clusters. Similarly, the  $\Sigma \gamma$  values of the (SA)<sub>1</sub>·(A)<sub>3</sub>·(LAS)<sub>2</sub> 252  $(1.99 \times 10^{0} \text{ s}^{-1})$  and  $(SA)_{2} \cdot (A)_{3} \cdot (LAS)_{1} \cdot (2.29 \times 10^{-4} \text{ s}^{-1})$  clusters at 278.15 K were found to be  $10^{1}$ -253 10<sup>5</sup> times lower than the most stable binary cluster, (SA)<sub>3</sub>·(A)<sub>3</sub> (3.28 × 10<sup>1</sup> s<sup>-1</sup>). Moreover, these 254 255 clusters exhibited  $\beta C/\sum \gamma$  ratios greater than 1 (Table S11), suggesting a favorable balance between 256 cluster growth and evaporation. Similar trends in  $\Delta G$  and  $\Sigma \gamma$  were observed across the other temperatures studied, including 238.15 K, 258.15 K and 298.15 K. Taken together, the  $\Delta G$  and  $\Sigma \gamma$ 257 258 analyses provide strong evidence that LAS incorporation enhances SA-A cluster stability, thereby

260

261262

263

266267

increasing their likelihood of participating in nucleation events.

## 3.2.2 Cluster formation pathways

ACDC simulation were conducted at 278.15 K, with the concentrations of [SA] (10<sup>6</sup> molecules cm 3), [A] (10<sup>9</sup> molecules cm<sup>-3</sup>) and [LAS] (10<sup>5</sup> molecules cm<sup>-3</sup>). The results are presented in Fig. 5(a), illustrating two distinct mechanisms for cluster growth. The first pathway (depicted by black arrows) corresponds to pure SA-A clustering, starting from the (SA)<sub>1</sub> (A)<sub>1</sub> dimer. Subsequent stepwise addition of SA or A monomers drives the assembly of progressively larger and more stable clusters such as (SA)<sub>3</sub> (A)<sub>3</sub>, which eventually exits the system. The second pathway (depicted by blue arrows) includes clusters containing LAS, in which LAS performs two distinct roles: one as a "catalyst" and the other as a "participant". When LAS acts as a "catalyst", the (SA)<sub>1</sub> (A)<sub>2</sub> (LAS)<sub>1</sub> trimer collides with the SA monomer, forming the (SA)<sub>2</sub>·(A)<sub>2</sub>·(LAS)<sub>1</sub> cluster. Subsequently, LAS evaporates from the cluster, leaving behind the (SA)<sub>2</sub>·(A)<sub>2</sub> cluster. Meanwhile, when LAS acts as a "participant", collisions between the (SA)1 (A)1 dimer and LAS monomers lead to the assembly of the (SA)<sub>1</sub>·(A)<sub>1</sub>·(LAS)<sub>1</sub> cluster. This trimer then undergoes further collisions with either an SA or A monomer, producing the (SA)<sub>2</sub>·(A)<sub>3</sub>·(LAS)<sub>1</sub> cluster, which ultimately grows out of the system. These dual roles of LAS in SA-A clusters are observed across other temperatures of 298.15 K, 238.15 K and 258.15 K; however, at lower temperatures, such as 238.15 K, the LAS-involved pathway simplifies (as shown in Figs. S8, S9 and S10). As shown in Fig. 5(b), the contributions of LAS to the SA-A nucleation process was examined across a range of temperatures, with a focus on the nucleation mechanism that involves LAS participation. As temperature increases, the influence of LAS-involved pathways becomes progressively more dominant. At lower temperatures (238.15 and 258.15 K), SA-A clustering remains the dominant process, accounting for 73% of nucleation events, while LAS-involved pathways contribute a modest 21%. However, as the temperature rises to 278.15 K, LAS participation increases to 33%, signaling a more prominent role in cluster growth. At 298.15 K, this contribution further rises to 49%, nearly double that observed at the lower temperatures. These results highlight the crucial role of elevated temperatures in enhancing LAS's contribution to SA-A nucleation, emphasizing the temperature-dependent amplification of LAS-driven cluster formation.

To investigate the detailed nucleation pathways of LAS in the formation of SA-A clusters,

#### 3.2.3 Atmospheric implications of LAS

289 In addition to temperature, the concentrations of precursors play a pivotal role in SA-A aerosol 290 nucleation. Atmospheric LAS concentrations exhibit considerable variability across different global 291 environments (Tan et al., 2022; Mochizuki et al., 2017; Ristovski et al., 2010; Hettiyadura et al., 292 2017; Kanellopoulos et al., 2022). For example, lower LAS concentrations, ranging from  $1.00 \times 10^4$ 293 to  $8.34 \times 10^5$  molecules cm<sup>-3</sup>, are found in regions such as eucalypt forest (Ristovski et al., 2010), 294 Mt. Tai (China) (Mochizuki et al., 2017) and Athens (Kanellopoulos et al., 2022). In contrast, higher 295 LAS concentrations have been recorded in Centreville, Alabama (1.77 × 10<sup>6</sup> molecules cm<sup>-3</sup>) 296 (Hettiyadura et al., 2017), with peak levels in Patra (Kanellopoulos et al., 2022), reaching up to  $1.70 \times 10^7$  molecules cm<sup>-3</sup>. Similarly, the concentrations of SA and A vary, with SA ranging from 297 298 10<sup>4</sup>-10<sup>7</sup> molecules cm<sup>-3</sup> (Zhang et al., 2024; Ding et al., 2019), and A ranging from 10<sup>7</sup>-10<sup>11</sup> 299 molecules cm<sup>-3</sup> (Wu et al., 2017; Luo et al., 2014). Elevated concentrations of these species are 300 particularly prominent in regions such as northern China, the Midwestern United States, and 301 agricultural areas in Europe. Based on field observations of LAS, SA and A concentrations, the 302 contribution of LAS to SA-A nucleation was systematically assessed. As illustrated in Fig. S11, the 303 impact of LAS on the SA-A system is primarily governed by the concentrations of LAS and SA, 304 with minimal dependence on [A]. Consequently, Fig. 6 illustrates how the contribution ratio of LAS 305 varies with different concentrations of SA and LAS, under the previously identified favorable high-306 temperature condition of 278.15 K. 307 The three pie charts in the upper map illustrate the changing contribution of LAS to SA-A 308 aerosol nucleation as SA concentration increases from  $3.00 \times 10^4$  to  $6.00 \times 10^4$  molecules cm<sup>-3</sup>, with 309 a corresponding decrease in LAS contribution as [SA] rises. In regions characterized by low SA 310 concentrations (3.00 × 10<sup>4</sup> molecules cm<sup>-3</sup>), such as Hyytiälä, nucleation is predominantly driven 311 by the LAS-SA-A pathway, contributing approximately 93%. However, at higher SA concentrations (up to  $2.00 \times 10^6$  molecules cm<sup>-3</sup>), such as on the west coast of Ireland (O'dowd et al., 2002), the 312 LAS contribution drops from 93% to 33%. At even higher SA levels (up to  $1.00 \times 10^7$  molecules cm<sup>-</sup> 313 3), LAS-involved pathways account for only 18% of the total nucleation flux, as observed in Beijing, 314 315 China (Wang et al., 2011). These findings highlight that lower [SA] levels substantially amplify the 316 contribution of LAS contribution to SA-A aerosol nucleation. 317 The contribution of LAS to SA-A aerosol nucleation increases with LAS concentration, ranging from  $1.00 \times 10^4$  to  $1.77 \times 10^6$  molecules cm<sup>-3</sup>, as shown in the pie chart below the map. 318

319 This pattern indicates a positive correlation between LAS concentration and its contribution to 320 nucleation. In regions with low LAS concentrations  $(1.00 \times 10^4 \text{ molecules cm}^3)$ , such as eucalypt forests (Ristovski et al., 2010), LAS-mediated pathways account for only 15% of the total nucleation 321 322 flux. In areas with moderate LAS concentrations, such as Athens (8.34 × 10<sup>5</sup> molecules cm<sup>-3</sup>) 323 (Kanellopoulos et al., 2022) and Mt. Tai (1.00 × 10<sup>5</sup> molecules · cm<sup>-3</sup>) (Mochizuki et al., 2017), LAS 324 contribution increases substantially, rising from 15% to 73%. At high [LAS], as observed in the 325 Centreville, Alabama (1.77 × 10<sup>6</sup> molecules cm<sup>-3</sup>) (Hettiyadura et al., 2017), LAS-driven nucleation 326 becomes dominate, contributing up to 97 % of the total nucleation rate. These findings underscore 327 that elevated LAS concentrations significantly enhance SA-A nucleation. Thus, in regions 328 characterized by high T, low [SA], high [A] and high [LAS], especially in agricultural-developed 329 areas and forested areas, the LAS contribution to SA-A aerosol nucleation can be substantial. 330 3.3 The comparsion of enhancement effect between LAS and LA To evaluate the relative enhancing effects of LA versus LAS in the typical SA-A-driven 331 332 nucleation. The  $\Delta G$  (pink histograms) and  $\sum \gamma$  (red points) of the  $(LAS)_x(SA)_y(A)_3$  and 333  $(LA)_x(SA)_y(A)_3$  (x = 0 - 3, x + y = 3) clusters at 278.15 K are presented in Fig. 7(a) as a comparison. 334 The (SA)<sub>3</sub>(A)<sub>3</sub> cluster, the thermodynamic minimum of the SA-A system (Chen et al., 2025; Li et 335 al., 2020a), was chosen as a reference for comparison. Relative to this baseline, (LA)<sub>1-3</sub>(SA)<sub>0-2</sub>(A)<sub>3</sub> 336 clusters consistently exhibited higher  $\Delta G$  values, elevated by roughly 18.36-41.94 kcal mol<sup>-1</sup>. In 337 contrast, (LAS)<sub>1-3</sub>(SA)<sub>0-2</sub>(A)<sub>3</sub> clusters were slightly more stable, differing from the reference by only 338 0.09-5.80 kcal·mol<sup>-1</sup>. This suggests that LAS incorporation leads to a slight stabilization of the 339 cluster relative to LA. 340 Moreover, the evaporation rate  $(\sum \gamma)$  of the  $(Org)_x(SA)_y(A)_3$  (Org = LA and LAS; x = 1-3, x +341 y=3) clusters do not exhibit a simple relationship with the proportion of organic components within the clusters. The highest  $\sum \gamma$  was observed for the  $(Org)_2 \cdot (SA)_1 \cdot (A)_3$  (Org = LA and LAS) clusters, 342 343 regardless of whether LA or LAS was used. For the (LAS)<sub>1</sub>·(SA)<sub>2</sub>·(A)<sub>3</sub> and (LAS)<sub>3</sub>·(A)<sub>3</sub> clusters, the  $\Sigma \gamma$  ranged from  $10^{-4}$  to  $10^{-1}$  s<sup>-1</sup>, lower than that of the (SA)<sub>3</sub>·(A)<sub>3</sub> cluster, indicating that replacing 344 345 one or three SA molecules with LAS enhances the thermodynamic stability of the clusters. In 346 contrast, the  $\sum \gamma$  of the LA-SA-A clusters were found to be higher than those of the corresponding 347 LAS-SA-A and (SA)<sub>3</sub>·(A)<sub>3</sub> clusters, as displayed in Fig. 7(a). The LAS-SA-A clusters exhibit more 348 negative negative  $\Delta G$  values and lower  $\Sigma \gamma$ , suggesting that their formation is thermodynamically

349 more favorable than that of the LA-SA-A system. This enhanced stability can be attributed to 350 stronger interactions between LAS and SA-A systems relative to those between LA and SA-A. Based on these results, we can conclude that LAS, produced through the LA + SO<sub>3</sub> reaction, more 351 352 effectively stabilizes the SA-A system than LA itself. 353 Fig. 7(b) illustrates the variation in the cluster formation rate (J) and enhancement strength (R) 354 as a function of [LAS] and [LA] at 278.15 K, under the condition of [SA] = 106 molecules cm<sup>-3</sup> and 355 [A]= $10^9$  molecules cm<sup>-3</sup>. In the LAS-SA-A system, J increases sharply with rising [LAS], particularly when [LAS] exceeds 10<sup>5</sup> molecules cm<sup>-3</sup>. As [LAS] grows from 10<sup>5</sup> to 10<sup>6</sup> 356 357 molecules cm<sup>-3</sup>, J for the LAS-SA-A system rises by three orders of magnitude, whereas in the LAS-358 SA-A system, J exhibits only a modest increase from  $3.36 \times 10^{-9}$  to  $1.12 \times 10^{-8}$  cm<sup>-3</sup> s<sup>-1</sup>, consistent 359 with the corresponding increase in [LA] (Fig. 7b). Although LAS concentrations are typically three 360 orders of magnitude lower than LA (Tan et al., 2022), LAS exerts a substantially stronger enhancement effect in SA-A-driven nucleation. These contrasting trends are primarily due to the 361 362 combined influence of cluster thermodynamic properties  $\Delta G$  and  $\Sigma \chi$ , and the concentrations of 363 organic species within the respective systems. The sharp increase in J for the LAS-SA-A system 364 stems from the favorable  $\Delta G$  and low  $\sum \gamma$  of the (LAS)<sub>x</sub>(SA)<sub>y</sub>(A)<sub>z</sub> clusters, along with the relatively 365 high non-equilibrium concentration of LAS. In contrast, the less favorable  $\Delta G$  and higher  $\sum \gamma$  of the  $(LA)_x(SA)_y(A)_z$  clusters limit the kinetic efficiency of the LA-SA-A system, even at elevated [LA]. 366 367 This study reveals that the reaction of LA and SO3 generates LAS which acts as an effective 368 atmospheric nucleation precursor and significantly accelerates SA-A nucleation. Consequently, 369 atmospheric LA can react with part of SO<sub>3</sub>, potentially accounting for the relatively low observed 370 low SA concentration, while the generated LAS markedly promotes SA-A-driven NPF under such 371 conditions. To date, the effects of hydroxy acids and their derivatives on atmospheric NPF have not 372 been comprehensively investigated. The mechanism proposed here offers a general approach to 373 evaluate the roles of these acids, like 2-methylglyceric acid, aromatic acids and their derivates, 374 influence atmospheric nucleation processes. Incorporating this novel OSs pathways into 375 contemporary atmospheric models will advance the quantitative understanding of OSs' 376 contributions to aerosol formation. Furthermore, OSs originating from secondary processes, such as 377 gas-phase chemical reactions, deserve further observation and evaluation.

## 4 Conclusions

379 In this study, quantum chemical calculations, master equation analysis, and the ACDC kinetic 380 model were employed to investigate the cycloaddition reaction between SO<sub>3</sub> and LA, the role of 381 LAS in SA-A nucleation, and its impact on NPF. 382 Quantum chemical results in the gas phase indicate that SA and H2O effectively lower the 383 reaction barriers for LAS formation from the LA-SO<sub>3</sub> reaction, functioning as catalysts and even 384 enabling a barrierless reaction. The effective rate coefficient for the SO<sub>3</sub>-LA reaction catalyzed by 385 SA (10<sup>7</sup> molecules cm<sup>-3</sup>) is 4-5 times higher than the pathway catalyzed by H<sub>2</sub>O (10<sup>17</sup> molecules cm<sup>-3</sup> 386 3), making it more effective, particularly at altitudes of 5-10 km. Additionally, the effective rate 387 coefficients for LA ( $10^{12}$  molecules cm<sup>-3</sup>) catalyzing the SO<sub>3</sub> + H<sub>2</sub>O  $\rightarrow$  SA reaction is about  $10^{1}$ - $10^{4}$ 388 times larger than the corresponding values for SO<sub>3</sub> hydrolysis catalyzed by H<sub>2</sub>SO<sub>4</sub> (10<sup>7</sup> molecules·cm<sup>-3</sup>), HNO<sub>3</sub> (10<sup>9</sup> molecules·cm<sup>-3</sup>), HCOOH (10<sup>11</sup> molecules·cm<sup>-3</sup>), and OA (10<sup>9</sup> 389 390 molecules cm<sup>-3</sup>), indicating that LA primarily catalyzes SO<sub>3</sub> hydrolysis within the temperature 391 range of 280-320 K. 392 LAS, functioning as both a hydrogen-bond donor and acceptor, participates in SA-A-driven 393 ternary nucleation, directly interacting with SA and A. Gibbs free energy analysis demonstrates that 394 ternary SA-A-LAS clusters consistently exhibit lower  $\Delta G$  values than their binary counterparts, 395 suggesting that LAS incorporation stabilizes the SA-A clusters. ACDC kinetic simulations further 396 demonstrate that LAS significantly enhances NPF, especially at low temperatures, low SA 397 concentration, and high A and LAS concentrations. In regions with elevated LAS concentrations, 398 such as Centreville, Alabama, particle formation rates can increase by up to 108-fold, with SA-A-399 LAS clusters contributing up to 97% of the overall cluster formation pathways. It is noteworthy that 400 LAS not only acts as a catalyst in enhancing SA-A cluster stability but also directly participates in 401 nucleation. Moreover, LAS exerts a stronger enhancement effect than LA, making it a more 402 effective stabilizing agent for atmospheric NPF. These findings suggest that LAS plays a critical 403 role in enhancing SA-A-driven NPF in forested and agriculturally developed regions, providing 404 insights into previously unaccounted NPF sources and refining nucleation models. 405 This study deepens the understanding of OSs formation in organic acid-polluted regions and 406 underscores the potential contribution of other OSs to NPF. Neglecting the contribution of OSs in

- the SA-A aerosol nucleation, particularly in forested and agricultural regions, may lead to an
- underestimation of organic aerosol nucleation risks.

## Acknowledgments

- This work was supported by the National Natural Science Foundation of China (No: 22203052;
- 22073059) and the Funds of Graduate Innovation of Shaanxi University of Technology (No:
- SLGYCX2506).

409

### 413 Declaration of competing interest

- The authors declare that they have no known competing financial interests or personal
- relationships that could have appeared to influence the work reported in this paper.

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

Graphic abstract

Fig. 1 Potential energy profiles and corresponding molecular structures for the LA + SO<sub>3</sub>  $\rightarrow$  LAS reaction in the absence and presence of H<sub>2</sub>O and H<sub>2</sub>SO<sub>4</sub> investigated at the CCSD(T)-F12/cc-pVDZ-F12//M06-2X/6-311++G(2df, 2pd) level

Fig. 2(a) Effective rate constants for the LA + SO<sub>3</sub>  $\rightarrow$  LAS reaction in the presence of H<sub>2</sub>O ( $k'_{WM}$ , cm<sup>3</sup>·molecule<sup>-1</sup>·s<sup>-1</sup>) and H<sub>2</sub>SO<sub>4</sub> ( $k'_{SA}$ , cm<sup>3</sup>·molecule<sup>-1</sup>·s<sup>-1</sup>) calculated using the master equation over the temperature range of 230-320 K; (b) Effective rate constants (k', s<sup>-1</sup>) for the hydrolysis of SO<sub>3</sub> with various species X(X = LA, SA, NA, FA and OA) within the temperature range of 230-320 K, where SA, NA, FA and OA are denoted as H<sub>2</sub>SO<sub>4</sub>, HNO<sub>3</sub>, HCOOH and H<sub>2</sub>C<sub>2</sub>O<sub>4</sub>, respectively.

727728

729

**Fig. 3** Electrostatic potential (ESP)-mapped van der Waals surfaces of A, LAS and SA molecules. ESP minima and maxima for different functional groups are shown as blue and yellow spheres, respectively, with their corresponding values (kcal·mol<sup>-1</sup>) indicated in parentheses. Red arrows denote preferred directions for hydrogen bond formation, while blue arrows illustrate likely pathways for proton transfer.

733

734

**Fig. 4** The total evaporation rates  $(\Sigma \gamma)$  (s<sup>-1</sup>) of  $(SA)_x(A)_y(LAS)_z$  ( $y \le x + z \le 3$ ) clusters at 278.15 K and 1 atm calculated at the M06-2X/6-311++G(2*df*, 2*pd*) level of theory. (a) without LAS monomer, (b) containing 1 LAS monomer, (c) containing 2 LAS monomers, and (d) containing 3 LAS monomers

737

738739

**Fig. 5** Nucleation mechanism of the LAS-SA-A system. (a) Cluster formation pathway at 278.15 K, with concentrations of [SA] =  $10^6$ , [A] =  $10^9$  and [LAS] =  $10^5$  molecules cm<sup>-3</sup>; (b) the branch ratio of outward flux at different temperatures. Only net fluxes contributing more than 5% to cluster growth are depicted.

743

**Fig. 6** Branching ratios of SA-A-LAS (red) and SA-A (blue) cluster growth pathways in regions with varying [LAS] concentrations. Black data points indicate field observations, while blue points represent the median values used in this study. Ammonia concentration is fixed at 10<sup>9</sup> molecules cm<sup>3</sup>. Map source: ©Google Maps (https://www.google.com/maps)

Fig. 7 (a) Gibbs free energies  $\Delta G$  (kcal·mol<sup>-1</sup>) and total evaporation rates  $\sum \gamma$  (s<sup>-1</sup>) for  $(LA)_x(SA)_y(A)_3$  and  $(LAS)_x(SA)_y(A)_3$  (x = 0.3, x + y = 3) clusters calculated at the M06-2X/6-311++G(2df, 2pd) level of theory and 278.15 K; (b) Cluster formation rate (J) and enhancement strength (R) for LAS as a function of monomer concentrations ([LA] and [LAS]) at 278.15 K, with [SA] fixed at  $10^5$  molecules·cm<sup>-3</sup> and [A] at  $10^9$  molecules·cm<sup>-3</sup>.