# Peer review of "Unexpected enhancement of new particle formation by lactic acid"

_EGUsphere, 2025_

## Author Comment (AC3)

**Responses to Referee #1's comments**

We are grateful to the reviewers for their valuable and helpful comments on our manuscript "Unexpected enhancement of new particle formation by lactic acid sulfate resulting from $SO_3$ loss in forested and agricultural regions" (Manuscript ID: egusphere-2025-4894). We have revised the manuscript carefully according to reviewers' comments. The point-to-point responses to the Referee #1's comments are summarized below:

**Referee Comments:**

The manuscript by Wang et al. presents a comprehensive theoretical investigation into the formation mechanism of lactic acid sulfate (LAS) and its unexpected role in enhancing sulfuric acid-ammonia (SA-A) driven new particle formation (NPF). The combination of quantum chemical calculations and ACDC kinetic modeling provides molecular-level insights into the catalytic effects of SA and $H_2O$ on LAS formation and the role of LAS in enhancing SA-A nucleation. This study advances our molecular-level mechanistic understanding of how organosulfates influence nucleation events. The manuscript is well-structured and clearly written. Therefore, I recommend publication of this manuscript after consideration of the following comments.

**Response:** We would like to thank the reviewer for the positive and valuable comments, and we have revised our manuscript accordingly.

**Specific Comments:**

**Comment 1:**

In the Introduction (Lines 52-53), the rationale for specifically investigating the enhancement of the SA-A system is introduced somewhat abruptly. The transition to this focus would be strengthened by briefly outlining the existing evidence or theoretical basis that suggests such an enhancement is plausible and significant.

**Response:** Thank you for your valuable comments. To better transition to the focus on the ternary nucleation process driven by SA-A, relevant existing studies and theoretical foundations have been added in the introduction, emphasizing the plausibility and significance of this enhancement effect, thereby providing a clearer rationale for investigating the enhancement of the SA-A system. According to the reviewer's suggestion, in Lines 52-55 Page 2 of the revised manuscript, the

sentence of "In response to this, several studies have explored the role of ternary nucleation driven by SA-A, which involves a broader array of atmospheric species, including ammonia ($NH_3$) (Li et al., 2020b; Yin et al., 2021a), organic amines (Li et al., 2017; Li et al., 2018a), organic and inorganic acids (Wang et al., 2011; Liu et al., 2018), and highly oxidized multifunctional compounds (HOMs) (Liu et al., 2021a; Liu et al., 2019a; Ning and Zhang, 2022; Yin et al., 2021b; Zhang et al., 2018)." has been changed as "Then plenty of low weight molecular organic acids such as glycolic acid (Zhang et al., 2017), malonic acid (Zhang et al., 2018) and pyruvic acid (Tsona Tchinda et al., 2022) also exhibit enhancement effects on ternary nucleation driven by SA-A nucleation system through catalytic mechanisms.".

**Comment 2:**

The acronym "SA-A" (where "A" stands for ammonia) is inconsistent with the use of "$NH_3$" throughout the text. This can be confusing for readers. For improved readability, please adopt a single, consistent acronym.

**Response:** Thanks for the suggestion of the reviewer. We apologize for the misunderstanding about ammonia. As the suggestion of the reviewer, the name of ammonia have been corrected. Specifically, ammonia has been labeled as "ammonia (A)" when they are first used. Besides, when they are used again, ammonia has been labeled as "A". The specific revisions are as follows:

(a) In Line 45 Page 2 of the revised manuscript, the "ammonia ($NH_3$)" has been changed as "ammonia (A)".

(b) In Line 82 Page 3 of the revised manuscript, the "ammonia" has been changed as "A".

**Comment 3:**

In Section 2.1, regarding the search for the global minimum configuration of the $(SA)_x(A)_y(LAS)_z$ clusters (when $y = 3$, $x + z = 3$), it is unclear whether the sampling of 4000 initial configurations is sufficient to adequately explore the complex conformational space.

**Response:** Thanks for your valuable comments. Indeed, a multi-path searching approach is utilized to explore the stable structures of $(LAS)_x(SA)_y(A)_z$ ( where $0 \leq y \leq x + z \leq 3$). For each global minimum cluster of $(LAS)_x(SA)_y(A)_z$ ( where $0 \leq y \leq x + z \leq 3$), $n$ different searching pathways were considered to ensure a thorough exploration of the complex conformational space. Specifically, a single monomer is incorporated to form a larger cluster on top of the existing smaller ones. For instance, in the process of searching for the stable structure of $(SA)_2 \cdot (A)$ clusters, two search

pathways exist: $(SA)\cdot(A) + SA$ and $(SA)_2 + A$. Similarly, in the search for the stable structure of $(LAS)\cdot(SA)\cdot(A)$ clusters, three pathways are considered: $(SA)\cdot(A) + LAS$, $(SA)\cdot(LAS) + A$ and $(LAS)\cdot(A) + SA$. Additionally, we apologize for the incorrect range of $n$ values previously used. Upon reviewing all the search pathways, we confirm that the correct range for $n$ is $1 \leq n \leq 3$, rather than n = 2 to 4. Consequently, the sentence of "a diverse set of initial structures $n \times 1000$ ($1 < n \leq 4$) were randomly produced." has been changed as "a diverse set of initial structures $n \times 1000$ ($1 \leq n \leq 3$) were randomly produced." in Line 115 on Page 5 of the revised manuscript.

**Comment 4:**

The computational details, such as the definition of boundary clusters and the coagulation sink in the ACDC simulations, should be more thoroughly described in the main text or supplementary information to ensure reproducibility.

**Response:** Thank you for your valuable comments. According to your suggestion, boundary conditions and the coagulation sink in the ACDC simulations have been added in Lines 155-160 Page 6 of the revised manuscript, which has been organized as "Sensitivity tests were conducted by varying the condensation sink (Cs) from $6 \times 10^{-4} \sim 6 \times 10^{-2}$ s$^{-1}$, indicating that the Cs exerted minimal influence on the main conclusions (Fig. S11). Therefore, the Cs was set to a representative value of $2.6 \times 10^{-3}$ for all subsequent calculations (Liu et al., 2021). Additionally, $(LAS)_4(A)_3$, $(LAS)_4(A)_4$, $(LAS)_2(SA)_2(A)_3$, $(LAS)_2(SA)_2(A)_4$, $(LAS)(SA)_3(A)_3$, $(LAS)(SA)_3(A)_4$, $(SA)_4(A)_3$ and $(SA)_4(A)_4$ clusters are acting as boundary clusters for LAS-SA-A system.".

**Comment 5:**

Line 210, please explain the reason for introducing lactic acid sulfuric anhydride (LASA). A clarification on its chemical relationship and distinction to LAS would be helpful for readers to follow the viewpoint.

**Response:** Thank you for your valuable comments. Following your suggestion, the distinctions between lactic acid sulfate (LAS) and lactic acid sulfuric anhydride (LASA) have been clarified. Both LAS and LASA are products of the reaction between $SO_3$ and lactic acid (LA). LAS is generated via esterification of the hydroxyl group of LA with $SO_3$, whereas LASA is formed through cycloaddition of the carboxyl group of LA, as illustrated by the potential energy surfaces in Fig. S1. Based on this, the detailed introduction of LASA has been clarified and added in Lines 215-216 Page 8 of the revised manuscript, which has been organized as "LASA, the product from the reaction

between $SO_3$ and the carboxyl group of LA, Fig. S1".

**Comment 6:**

In Section 3.2.3, please clarify how "LAS contribution" is quantitatively calculated. Specifically, is it determined by the fraction of outgrowing clusters that contain at least one LAS molecule? A brief description of the calculation methodology is needed.

**Response:** Thank you for your valuable comments. According to your suggestion, regarding the calculation of the final outgoing fluxes of LAS in the ACDC simulations have been added in Lines 160-161 Page 6 of the revised manuscript, which has been organized as "Also, the details of the contribution of LAS to SA-A nucleation was estimated in the first part of the Supplement".

To further elaborate on the computational procedures, we have included the specific outgoing fluxes of LAS in the ACDC simulations in the supplementary material, organized as follows: "To quantify the contribution of LAS to SA-A nucleation, we analyzed the steady-state cluster formation fluxes ($J$) output by ACDC. The LAS-related nucleation fraction at a given temperature was defined as the ratio of the total formation flux of all nucleated clusters containing at least one LAS molecule to the total nucleation flux from all pathways. Specifically,

$$f_{LAS}(T) = \frac{\sum J_{out}(C_i | C_i \in C_{LAS})}{\sum J_{out}(C_i)} \tag{S1}$$

where $J_{out}(C_i)$ denotes the outgoing flux of cluster $C_i$ that meets the nucleation criterion (size/composition threshold), and $C_{LAS}$ represents the set of all nucleated clusters containing one or more LAS molecules. This metric directly reflects the proportion of new particles formed through LAS-involved pathways under steady-state conditions." .

**Comment 7:**

In Section 3.2.3, the authors state that LAS contributions are more pronounced in forested and agricultural regions. It would be helpful to clarify why these regions exhibit higher LAS relevance compared to urban or industrial areas.

**Response:** Thank you for your valuable comments. The influence of LAS on sulfuric acid-ammonia (SA-A)-driven new particle formation (NPF) show that, under elevated SA concentrations, such as those commonly observed in urban or industrial environments, the dominant effect of SA suppresses the nucleation-promoting role of LAS, thereby substantially diminishing its contribution to nucleation. In contrast, in forested and agricultural regions characterized by relatively low SA

concentrations and more abundant sources of LAS, LAS exhibits a markedly stronger nucleation-promoting effect within the sulfuric SA-A system. Accordingly, we infer that in forested and agricultural regions, which are typically characterized by low SA concentrations, LAS plays a pronounced role in promoting SA-A nucleation. The corresponding explanations can be mainly attributed to two aspects.

(a) The enhancement factor $R$ in the SA-A-based system is strongly influenced by the concentrations of both SA and LAS. To illustrate this dependence, Fig. S10 shows $R$ as functions of [SA] and [LAS] under the conditions of [A] = $10^9$ molecules·cm$^{-3}$ and $T$ = 278.15 K. As depicted in Fig. S10, the $R$ increases as the [SA] decreases. At low [SA] ($10^5$ molecules·cm$^{-3}$), the $R$ value reaches its maximum of $1.32 \times 10^7$ fold at [LAS] = $10^6$ molecules·cm$^{-3}$. In contrast, at high [SA] ($10^7$ molecules·cm$^{-3}$), the influence of [LAS] on $R$ is markedly reduced, resulting in a 1.54 fold increase in $R$ at [LAS] = $10^6$ molecules·cm$^{-3}$. Based on the above analysis, in Lines 322-324 Page 12 of the revised manuscript, the discussion of why agricultural-developed and forested areas exhibit higher LAS relevance compared to urban or industrial areas, which has been added and organized as "In contrast, environments with typically high SA concentrations, such as urban and industrial areas, promote SA-A self-aggregation nucleation, thereby diminishing the relative contribution of LAS (Fig. S10)."

[Figure]

**Fig. S10** Enhancement factor $R$ as functions of [SA] and [LAS] at [A] = $10^9$ molecules·cm$^{-3}$ and 278.15 K.

(b) During formation of LAS via the reaction of $SO_3$ with LA strongly competes with SA formation. To further evaluate the competitive interactions between LAS and SA molecules, another

set of ACDC simulations was conducted, considering different ratios of the concentrations of LAS and SA ([LAS]/[SA]) and varying the total concentration of LAS and SA. Fig. S12 shows particle formation rates ($J$, cm$^{-3}$·s$^{-1}$) with varying ratios of [LAS]:[SA] at 278.15 K under different concentrations ((a)$10^7$ molecules·cm$^{-3}$, (b)$10^9$ molecules·cm$^{-3}$ and (c)$10^{11}$ molecules·cm$^{-3}$). The specific revisions are as follows:

[Figure]

**Fig. S12** Particle formation rates ($J$, cm$^{-3}$·s$^{-1}$) with varying ratios of [LAS]:[SA] at 278.15 K under different concentrations ((a)$10^7$ molecules·cm$^{-3}$, (b)$10^9$ molecules·cm$^{-3}$, (c)$10^{11}$ molecules·cm$^{-3}$). [LAS] + [SA] = $10^4$-$10^8$

molecules·cm$^{-3}$

The corresponding revision has been added and organized as "The observed concentration dependence indicates that the LAS-driven nucleation process becomes particularly significant in environments with moderate to high LAS concentrations and relatively low SA levels. Therefore, in the LAS-SA-A ternary nucleation system, LAS is likely to function as an "acid" molecule, exhibiting a competitive effect. To evaluate the competitive interactions between LAS and SA molecules, another set of ACDC simulations was conducted, considering different ratios of the concentrations of LAS and SA ([LAS]/[SA]) and varying the total concentration of LAS and SA. Fig. S12 shows the variation of $J_{LAS/SA}$ with the total concentrations of SA and LAS at A concentrations of $10^7$, $10^9$, and $10^{11}$ molecules·cm$^{-3}$.

As shown in Fig. S12(a), at lower atmospheric concentration of A ($10^7$ molecules·cm$^{-3}$), the formation rate $J_{LAS/SA}$ increases with the substitution percentage. At a 50% substitution rate ([LAS]:[SA] = 1:1), $J_{LAS/SA}$ sharply increases to $1.46 \times 10^{-9}$ cm$^{-3}$·s$^{-1}$, which is larger by 1-2 orders of magnitude than the value at unsubstituted condition. At a 99% substitution rate ([LAS]:[SA] = 99:1), $J_{LAS/SA}$ reaches its maximum value of $6.99 \times 10^{-9}$ cm$^{-3}$·s$^{-1}$, which is 1-3 orders of magnitude greater than the value under non-substituted conditions. These results indicate that, at lower atmospheric concentrations of A, the enhancing effect of LAS on the SA-A group particle formation rate increases with the substitution percentage. At intermediate ($10^9$ molecules·cm$^{-3}$) and higher concentrations ($10^{11}$ molecules·cm$^{-3}$) of atmospheric A, the $J_{LAS/SA}$ at a 99% substitution rate ([LAS]:[SA] = 99:1) reaches its maximum value (Fig. S12(b) and Fig. S12(c)). Compared to the $J_{LAS/SA}$ under non-substituted conditions, the value at a 99% substitution rate is increased by one order of magnitude. In contrast, urban and industrial environments, which typically have high SA concentrations, favor SA-A self-aggregation nucleation, thereby reducing the relative contribution of LAS. Thus, in regions characterized by high $T$, low [SA], high [A] and high [LAS], especially in agricultural-developed areas and forested areas, the LAS contribution to SA-A aerosol nucleation can be substantial." in the supplementary material.

**Comment 8:**

Line 398 and Line 17: "particle formation rates can increase by up to $10^8$-fold", if this value is provided in the SI, please indicate where it can be found.

**Response:** Thanks for your valuable comments. We apologize for not clearly citing the source of

this data in the previous version of the manuscript. According to the reviewer's suggestion, the specific data indicating that particle formation rates can increase by up to $10^8$-fold has been clarified in the revised manuscript. The supporting data can be directly viewed in Fig. S5(b).

**Comment 9:**

Some minor mistakes are shown in the manuscript, e.g., Line 64: "PM10"; Line 100: "nucleation and particle formation (NPF)"; Line 267: "exits the system"; Line 348: "negative negative $\Delta G$ values"; Line 64: "whereas in the LAS-SA-A system" and so on. Please totally and carefully recheck the whole manuscript and correct all the mistakes.

**Response:** Thanks to the reviewer's insightful comment, we are sorry for the trouble we have caused by oversight. In order to improve the accuracy of the expression, the corresponding main revision has been made as follows:

(a) In Line 62 Page 3 of the revised manuscript, "PM10" has been changed as "$PM_{10}$".

(b) In Line 98 Page 4 of the revised manuscript, "nucleation and particle formation (NPF)" has been changed as "new particle formation (NPF)".

(c) In Line 272 Page 10 of the revised manuscript, "exits the system" has been changed as "exit the system".

(d) In Line 360 Page 13 of the revised manuscript, "negative negative $\Delta G$ values" has been changed as "negative $\Delta G$ values".

(f) In Line 369 Page 13 of the revised manuscript, "whereas in the LAS-SA-A system" has been changed as "whereas in the LA-SA-A system".

---

## Author Comment (AC4)

**Responses to Referee #2's comments**

We are grateful to the reviewers for their valuable and helpful comments on our manuscript "Unexpected enhancement of new particle formation by lactic acid sulfate resulting from $SO_3$ loss in forested and agricultural regions" (Manuscript ID: egusphere-2025-4894). We have revised the manuscript carefully according to reviewers' comments. The point-to-point responses to the Referee #2's comments are summarized below:

**Referee Comments:**

Wang et al. utilized quantum chemical calculations, master equation analysis, and Atmospheric Clusters Dynamic Code kinetic model to systematically investigate the formation mechanism of lactic acid sulfate (LAS) and its enhancing effect on sulfuric acid (SA)-$NH_3$(A) nucleation. Particular attention is given to the reaction between lactic acid and $SO_3$, the catalytic effects of $H_2O$/SA, and the dual role played by LAS in the SA-A-LAS ternary system (both as a participant and as a catalyst). The topic is novel, the methodology is sound, and the work provides an important-yet previously underappreciated-mechanistic explanation for the unusually high NPF rates observed in forested and agricultural regions. Most of this manuscript is well written and will be of broad interest to the readers of Atmospheric Chemistry and Physics. I recommend its publication in the journal, provided that the following comments are addressed.

**Response:** We would like to thank the reviewer for the positive and valuable comments, and we have revised our manuscript accordingly.

**Specific Comments**:

**Comment 1:**

The results indicate that the barriers to the reaction between lactic acid and $SO_3$ are substantially reduced with the addition of SA. However, the underlying mechanism driving SA's pronounced catalytic effect has not been adequately addressed. Providing one or two specific structural characteristics, such as the lengths of critical hydrogen bonds or specific geometric changes in transition states, would clarify why SA exhibits higher catalytic efficiency than $H_2O$, thereby allowing readers to fully comprehend the mechanism driving the "barrier reduction".

**Response:** Thank you for your valuable comments. According to your suggestion, the geometrical

structure of the eight-membered ring transition state $TS_{SA}$ has been compared with that of the six-membered ring transition state $TS_{WM}$. In Lines 199-202 Page 8 of the revised manuscript, this comparison is presented as follows: "As compared with six-membered ring transition state $TS_{WM}$, the transition state $TS_{SA}$ shows eight-membered ring structure, which reduces the ring tension greatly. So, from an energetic point of view, SA lowers the Gibbs free energy barrier to 3.5 kcal·mol$^{-1}$, 4.3 kcal·mol$^{-1}$ lower than the barrier observed for the $H_2O$-catalyzed pathway.".

**Comment 2:**

The authors' calculations reveal that the dominant nucleation pathways shift with temperature, however, the manuscript does not adequately explain why the contribution of LAS-related pathways increases with increasing temperature. Further clarification of the underlying mechanism is required, such as whether this behavior is associated with variations in collision frequency or the fact that LAS exhibits a relatively weak temperature dependence in its evaporation rate. Incorporating such an explanation would greatly enhance the interpretability of the trend presented in Fig. 5 of the manuscript.

**Response:** Thanks for your valuable comments. According to the reviewer's suggestion, the reason for the variation of LAS-related pathways with temperature has been added. The corresponding changes are as follows.

  (a) In Lines 285-287 Page 10 of the revised manuscript, the reason for the increasing influence of LAS-involved pathways with rising temperature is added and organized as " As temperature increases, the influence of LAS-involved pathways becomes progressively more dominant, due to the elevated vapor pressure of LAS raises its gas-phase concentration, thereby promoting further cluster formation.".

  (b) In Lines 288-290 Page 11 of the revised manuscript, at lower temperatures, the reason for the modest contribution of LAS-involved pathways is presented and organized as " At lower temperatures (238.15 and 258.15 K), SA-A clustering remains the dominant process, accounting for 73% of nucleation events, while LAS-involved pathways contribute a modest 21%, because of the reduced collision frequency of LAS.".

**Comment 3:**

The manuscript proposes that LAS may function either as a "participant" or as a "catalyst-like promoter," which is an interesting and meaningful finding. At present, the distinction between these

two roles is mainly inferred from the ACDC pathways in Fig. 5 (i.e., whether LAS ultimately remains in the cluster), whereas Fig. 6 and Fig. 7 primarily illustrate how the contribution of LAS varies with temperature and precursor concentrations. Their connection to the role distinction is not explicitly established. To make the origin of this "dual role" clearer, a brief clarification in the discussion section would help enhance the manuscript's logical coherence assist readers in better understanding how LAS behaves under different conditions.

**Response:** Thanks for your valuable comments. Specifically, we highlight that the dual role of LAS is determined by its behavior in the cluster formation pathway, as illustrated in Fig. 5. When LAS functions as a 'catalyst', it temporarily participates in the cluster formation but evaporates after facilitating the growth process. In contrast, when LAS acts as a 'participant', it remains within the cluster throughout the entire nucleation process. To make the origin of this 'dual role' clearer, the corresponding changes are as follows.

(a) In Lines 331-332 Page 12 of the revised manuscript, an example illustrating the role of LAS as a catalyst has been added and organized as "While LAS contributes to the initial stages of cluster formation, it subsequently evaporates from the pre-nucleation cluster, ultimately functioning in a catalyst-like capacity (Fig. S16).".

(b) In Lines 335-338 Page 12 of the revised manuscript, an example illustrating the role of LAS as a participant has been added and organized as "At high [LAS], as observed in the Centreville, Alabama ($1.77 \times 10^6$ molecules·cm$^{-3}$) (Hettiyadura et al., 2017), LAS-driven nucleation becomes dominant, resulting in a 'participant' synergistic nucleation mechanism that works like 'hand in hand' (Fig. S17), contributing up to 97 % of the total nucleation rate.".

**Comment 4:**

I suggest the authors explicitly outline how boundary conditions were set in their ACDC simulations, along with justifying the maximum cluster size they selected. Nucleation rates are often sensitive to the choice of boundary conditions. Accordingly, it is essential to clarify why setting the maximum cluster size at x + y + z ≤ 3 was adequate for their simulations, or alternatively, to discuss the implications of extending this boundary to larger clusters. Even a short, targeted explanation would greatly enhance the clarity and reproducibility of the methodology.

**Response:** Thanks for your valuable comments. For reviewers' comments, the corresponding revision has been made as follows.

(a) In ACDC simulations, boundary clusters are those allowed to flux out of the simulation box for further growth. Consequently, the smallest clusters outside the simulated system must be sufficiently stable to prevent immediate evaporation back into the system. In addition, considering the formation Gibbs free energy (Table S7) and evaporation rates (Table S9), the clusters $(LAS)_4(A)_3$, $(LAS)_4(A)_4$, $(LAS)_2(SA)_2(A)_3$, $(LAS)_2(SA)_2(A)_4$, $(LAS)(SA)_3(A)_3$, $(LAS)(SA)_3(A)_4$, $(SA)_4(A)_3$ and $(SA)_4(A)_4$ clusters are selected as the boundary clusters for LAS-SA-A system. Based on the above analysis, in Lines 158-160 Page 6 of the revised manuscript, the relevant information about boundary clusters have been added and organized as "Additionally, $(LAS)_4(A)_3$, $(LAS)_4(A)_4$, $(LAS)_2(SA)_2(A)_3$, $(LAS)_2(SA)_2(A)_4$, $(LAS)(SA)_3(A)_3$, $(LAS)(SA)_3(A)_4$, $(SA)_4(A)_3$ and $(SA)_4(A)_4$ clusters are acting as boundary clusters for LAS-SA-A system.".

(b) As reported by Besel et al. (*J. Phys. Chem. A*, 2020, 124(28), 5931-5943), the explicitly simulated set of clusters should always include the "critical cluster". Usually, the highest barrier on the lowest-energy path connecting the monomers to the outgrowing clusters (a saddle point on the actual $\Delta G$ surface) represents the "critical cluster". So, at 278.15 K (Fig. S4), the actual $\Delta G$ of $(A)_y(LAS)_z$ ($0 \leq y \leq z \leq 4$), $(SA)_x(A)_y$ ($0 \leq y \leq x \leq 3$), $(SA)_x(A)_y(LAS)_1$ ($0 \leq y \leq 4$, $0 \leq x \leq 3$), and $(SA)_x(A)_y(LAS)_2$ ($0 \leq y \leq 4$, $0 \leq x \leq 2$) clusters were calculated to verify that the $3 \times 3$ systems adequately capture the influence of LAS on SA-A nucleation. As seen in Fig. S4, the actual $\Delta G$ surface represented that the simulated set of clusters always included the critical cluster. So, we conclude that, in atmospherically relevant conditions, a $3 \times 3$ cluster set is adequate for predicting the particle formation in the SA-A system.

[Figure]

[Figure]

**Fig. S4** A typical actual Δ*G* surface at 278.15 K. [SA] is the concentration of sulfuric acid monomers, [A] the concentration of ammonia monomers and [LAS] is lactic acid sulfate

**Comment 5:**

A single value of $2.6 \times 10^{-3}$ s$^{-1}$ was adopted for the condensation sink in the ACDC kinetics simulation under different atmospheric conditions of agricultural and forested regions (Figure 6), without addressing whether this parameter is representative of such diverse conditions. In practice, condensation sinks can vary by orders of magnitude depending on aerosol loading. Hence, the manuscript ought to explain the rationale for using a single Cs value across all cases, or discuss the uncertainties associated with this choice for the cluster formation rates or pathways. Including such justification would greatly enhance the credibility of the modeled nucleation rates.

**Response:** We sincerely appreciate the reviewer's careful reading of our manuscript. As the reviewer pointed out, the condensation sink (Cs) coefficients vary significantly across regions. According to previous reports (Jayaratne et al., 2017; Qi et al., 2015; Shen et al., 2020), the effect of Cs on results was examined, by additional runs with various values covering cases of clean and haze days ($6 \times 10^{-4}$ to $6 \times 10^{-2}$ s$^{-1}$). To further evaluate the influence of Cs values on cluster formation rates, two additional sets of ACDC simulations were performed using different Cs values (Fig. S11). The results indicate that varying Cs value settings ($6 \times 10^{-4} \sim 6 \times 10^{-2}$ s$^{-1}$) does not affect the main conclusions of this study (Fig. S11). Thus, a representative Cs value of $2.6 \times 10^{-3}$ s$^{-1}$, was adopted as the sink term in the ACDC simulations. Following the reviewer's suggestion, in Lines 155-158 Page 6 of the revised manuscript, the sentence of "Here, the condensation sink coefficient was assigned $2.6 \times 10^{-3}$." has been changed as "Sensitivity tests were conducted by varying the

condensation sink (Cs) from $6 \times 10^{-4} \sim 6 \times 10^{-2}$ s$^{-1}$, indicating that the Cs exerted minimal influence on the main conclusions (Fig. S11). Therefore, the Cs was set to a representative value of $2.6 \times 10^{-3}$ for all subsequent calculations (Liu et al., 2021).".

[Figure]

**Fig. S11** The formation rate $J$ (cm$^{-3}$ s$^{-1}$) of LAS at varying consentrations of A and different condensation sink (Cs) values in the SA-A-LAS-based system where $T$ = 278.15 K, [SA] = $10^5$ molecules cm$^{-3}$, [LAS] = $10^3 \sim 10^6$ molecules cm$^{-3}$. Cs = $6 \times 10^{-4}$ s$^{-1}$ (dotted lines), $2.6 \times 10^{-3}$ s$^{-1}$ (solid lines) and $6 \times 10^{-2}$ s$^{-1}$ (dash-dotted lines)

**References**

Jayaratne, R., Pushpawela, B., He, C., Li, H., Gao, J., Chai, F., and Morawska, L.: Observations of particles at their formation sizes in Beijing, China, Atmos. Chem. Phys., 17, 8825-8835, 2017.

Qi, X. M., Ding, A. J., Nie, W., Petäjä, T., Kerminen, V. M., Herrmann, E., Xie, Y. N., Zheng, L. F., Manninen, H., Aalto, P., Sun, J. N., Xu, Z. N., Chi, X. G., Huang, X., Boy, M., Virkkula, A., Yang, X. Q., Fu, C. B., and Kulmala, M.: Aerosol size distribution and new particle formation in the western Yangtze River Delta of China: 2 years of measurements at the SORPES station, Atmos. Chem. Phys., 15, 12445-12464, 2015.

Shen, J., Elm, J., Xie, H. B., Chen, J., Niu, J., and Vehkamäki, H.: Structural effects of amines in enhancing methanesulfonic acid-driven new particle formation, Environ. Sci. Technol., 54, 13498-13508, 2020.

**Comment 6:**

Technical corrections:

Page 6 line 161: "In the direct cycloaddition pathway (Channel LAS) illsutrated in Fig. 1"

The word "illsutrated" should be corrected to "illustrated". In addition, there is a spelling error in the caption of Fig. 4, where "nunber" should be corrected to "number."

Page 5 line 114: "To identity the global minimum energy configurations of …"

The word "identity" should be corrected to "identify".

Page 10 line 278: "… the contributions of LAS to the SA-A nucleation process was examined, …"

The word "was" should be corrected to "were".

Page 12 lines 325-326: "LAS-driven nucleation becomes dominate, …"

The word "dominate" should be corrected to "dominant".

Page 21 lines 665-672: In the reference list, Yin et al., 2021a and Yin et al., 2021b share the same title and page numbers (Acid-base clusters during atmospheric new particle formation in urban Beijing" Environ. Sci. Technol., 55, 10994-11005). Please remove the duplicate references and update the citation numbers in the main text.

**Response:** Thanks to the reviewer's insightful comment, we are sorry for the trouble we have caused by oversight. In order to improve the accuracy of the expression, the corresponding main revision has been made as follows:

(a) In Line 164 Page 6 of the revised manuscript, "illsutrated" has been corrected to "illustrated".

(b) In Fig. 4, the "nunber" has been corrected to "number". The newly revised Fig. 4 is shown below.

[Figure]

**Fig. 4** The total evaporation rates ($\sum\gamma$) (s$^{-1}$) of $(SA)_x(A)_y(LAS)_z$ ($y \leq x + z \leq 3$) clusters at 278.15 K and 1 atm calculated at the M06-2X/6-311++G(2$df$, 2$pd$) level of theory. (a) without LAS monomer, (b) containing 1 LAS monomer, (c) containing 2 LAS monomers, and (d) containing 3 LAS monomers

(c) In Line 112 Page 4 of the revised manuscript, "identity" has been corrected to "identify".

(d) In Line 283 Page 10 of the revised manuscript, "was" has been corrected to "were".

(e) In Line 337 Page 12 of the revised manuscript, "dominate" has been corrected to "dominant".

(f) In Line 57 Page 3 of the revised manuscript, we have removed the duplicate reference and updated all corresponding citation numbers throughout the manuscript accordingly.